# Physical Restraint Use in Nursing Homes—Regional Variances and Ethical Considerations: A Scoping Review of Empirical Studies

**DOI:** 10.3390/healthcare11152204

**Published:** 2023-08-04

**Authors:** Gülendam Hakverdioğlu Yönt, Sezer Kisa, Daisy Michelle Princeton

**Affiliations:** 1Department of Nursing, Faculty of Health Sciences, Izmir Tinaztepe University, 35400 Izmir, Türkiye; gulendam.yont@tinaztepe.edu.tr; 2Department of Nursing and Health Promotion, Faculty of Nursing, OsloMet—Oslo Metropolitan University, 0130 Oslo, Norway; sezkis@oslomet.no

**Keywords:** elderly, falls, interventions, nursing homes, physical restraint, residents

## Abstract

Background: Physical restraints are known to violate human rights, yet their use persists in long-term care facilities. This study aimed to explore the prevalence, methods, and interventions related to physical restraint use among the elderly in nursing homes. Methods: The method described by Joanna Briggs was followed to conduct a scoping review without a quality assessment of the selected studies. An electronic search was conducted to find eligible empirical articles using MEDLINE, PsycINFO, EMBASE, Web of Science, Scopus, Google Scholar, CINAHL, and grey literature. The database search was performed using EndNote software (version X9, Clarivate Analytics), and the data were imported into Excel for analysis. Results: The prevalence of physical restraint use was found to be highest in Spain (84.9%) and lowest in the USA (1.9%). The most common device reported was bed rails, with the highest prevalence in Singapore (98%) and the lowest (4.7%) in Germany, followed by chair restraint (57%). The largest number of studies reported the prevention and/or risk of falls to be the main reason for using physical restraints, followed by behavioral problems such as wandering, verbal or physical agitation, and cognitive impairment. Most studies reported guideline- and/or theory-based multicomponent interventions consisting of the training and education of nursing home staff. Conclusions: This review provides valuable insights into the use of physical restraints among elderly residents in nursing homes. Despite efforts to minimize their use, physical restraints continue to be employed, particularly with elderly individuals who have cognitive impairments. Patient-related factors such as wandering, agitation, and cognitive impairment were identified as the second most common reasons for using physical restraints in this population. To address this issue, it is crucial to enhance the skills of nursing home staff, especially nurses, in providing safe and ethical care for elderly residents with cognitive and functional impairments, aggressive behaviors, and fall risks.

## 1. Introduction

The increasing number of elderly persons around the world represents a significant public health challenge for many countries. Recent data show that the ageing population is growing much faster than in the past [1]. By 2050, there may be 2.1 billion people aged 60+ and 426 million aged 80 and over [2]. Individuals aged 85 years and older are one of the fastest-growing segments of the population and their numbers are expected to increase to 19.4 million in the United States and 2.7 million in Canada by 2050 [3,4]. Indeed, the population aged 85 and older living in nursing homes has grown by 23.0% since 2011 in Canada [4]. Age-related impairment in physical, cognitive, and functional abilities leads to an increase in dependency among the elderly, primarily due to dementia and Alzheimer’s disease. According to a recent study, the number of people living with dementia is estimated to grow to 152.8 million by 2050 [1], and the number of people worldwide with Alzheimer’s disease will be 106.8 million by 2050 [5]. As a result, nursing home use is expected to increase, which will require public health planning and policies that address the long-term care needs of the elderly [5].

With greater numbers of elderly persons in nursing homes, restraint use is a frequently encountered challenge, affecting more than 50% of nursing home residents with five or six activities of daily living (ADL) limitations [6]. The term “restraint” is defined as “a device or medication that is used for the purpose of restricting the movement and/or behavior of a person” [7]. It is also frequently defined as restricting freedom of movement [8]. When considered this way, freedom-restraining devices can be seen as a form of violence [9]. Physical activity is essential for the functioning of older people [10,11]. A decrease in physical activity reduces functional abilities [12]. It increases reliance on caregivers, leading to the increased use of restrictive practices. Among these restrictive practices, physical restraint is often used to address responsive behaviors and prevent falls in places where residents are aged 65 and over and may have cognitive impairment [13,14]. Physical restraint is usually defined as “the use of physical force to prevent, restrict or subdue movement of a care recipient’s body, or part of a care recipient’s body, for the primary purpose of influencing the care recipient’s behavior” [15]. An accepted definition of physical restraint is “any action or procedure that prevents a person’s free body movement to a position of choice and/or normal access to his/her body by the use of any method, attached or adjacent to a person’s body that he/she cannot control or remove easily” [13]. Examples of physical restraint devices include vests, straps, limb ties, wheelchair bars and brakes, chairs that tip backwards, tightly tucked sheets, bed rails, belts in a chair, belts in bed, hand mitts, wrist restraints, table trays, and sleeping suits [13,16].

The prevalence of physical restraint use in nursing homes was reported to be 37% in Europe, 22% in North America (Robins et al., 2021), 31% in Canada [17], and 20% in Hong Kong. A recent scoping review reported that the physical restraint rate varied between 7.7% and 60.5% in European nursing homes [18]. Such a wide use of physical restraint needs to be reconsidered as it is associated with life-threatening clinical consequences, including head trauma, asphyxiation, and death, as well as legal and ethical concerns such as violating a person’s right to freedom and dignity [16,19,20]. Physically restrained residents had worse outcomes for behavioral issues, cognitive performance, falls, dependency on walking, daily living activities, contractures, urinary and fecal incontinence, deep vein thrombosis, and skin injuries [21,22,23]. Nursing home residents with a history of physical restraint were found to be at higher risk of experiencing cognitive decline [12,24,25], muscular atrophy, increased disorientation [25], decreased mobility, increased mortality [26] and of antipsychotic use [24]. Luo et al. (2020) found that trunk use increased the risk of fractures by almost three times among nursing home residents with Alzheimer’s disease/dementia [27]. Thus, elderly people should be protected from the excessive use of physical restraint [28].

The most common reasons given by nursing home staff for using physical restraint are safety, such as preventing falls or self-injury or harm to others, residents’ inappropriate behavior, such as agitation and wandering, the convenience of the staff, shortages of nurses, the complexity of care, high workloads, lack of knowledge about physical restraint, absence of person-centered care, and lack of legislation/guidelines [7,29,30,31,32,33]. However, empirical studies do not support the use of physical restraint. The evidence shows that a decrease in physical restraint use does not result in more falls or fall-related injuries [34,35,36]. Staff shortages was not a good excuse either: increasing the number of nursing staff did not lead to a reduction in physical restraint [37,38]. In addition, the number of nurses and doctors per patient, the adequacy of staff, and institutional features showed no correlation with physical restraint use [39].

Physical restraint use is recognized as a violation of human rights. Nurses are expected to respect and protect an individual’s autonomy, dignity, and rights [31]. Furthermore, international guidelines and recent studies suggest that a restraint-free nursing home and model of care with reasonable levels of safety is possible, and that physical restraint should not be used unless there is an immediate danger, such as severe imbalance [15,16,25,40]. Yet, physical restraints are still being used in long-term care facilities [8,13,29,41]. There is an increasing amount of research being conducted on the use of restraints in hospitals and intensive care units [42], and on nurses’ knowledge about alternatives to confining residents [28,42]. However, there is little information on the prevalence of and methods used in physical restraint, reasons for using restraints among elderly persons in nursing homes, and interventions to reduce restraint use. Thus, this review is significant in that it contributes to the current literature on physical restraint use in nursing homes by examining the prevalence, types of predictors associated with physical restraint use, and effective interventions used to reduce restraint among elderly nursing home residents. As the population ages, there is a growing need to understand restraint use in long-term care.

### 1.1. Objectives

This review aimed to map the methods and prevalence of physical restraint use among the elderly in nursing homes, systematically describe the reasons for using physical restraint, and map interventions and their effectiveness in obviating physical restraint use in nursing homes.

### 1.2. Research Questions

What are the prevalence rates and methods of physical restraint use among the elderly in nursing homes?What are the reasons for using physical restraint on the elderly in nursing homes?What are the critical gaps in the literature regarding interventions to obviate physical restraint use among the elderly in nursing homes?

## 2. Method

### 2.1. Design

This review mapped the relevant literature on physical restraint use among the elderly in nursing homes and identified key concepts, research gaps, and types and sources of evidence to inform practice, policymaking, and research. This review employed the following five steps developed by Arksey and O’Malley [43]:Identify the research question;Identify relevant studies;Select studies;Extract the data;Collate, summarize, and report the results.

### 2.2. Search Methods

The method described by Joanna Briggs was followed to conduct a scoping review without a quality assessment of the selected studies [44]. Keywords and MeSH terms were identified through an initial exploration of Web of Science and PubMed. The terms were reviewed and agreed upon by the research team and the librarian from Izmir Tinaztepe University. After agreeing on the search terms, an electronic search was conducted to find eligible empirical articles using MEDLINE, PsycINFO, EMBASE, Web of Science, Scopus, Google Scholar, CINAHL, and grey literature by the librarian. The term “physical restraint” covered all restraints and was not differentiated as with or without a bed rail (bilateral/unilateral). The search strategy included the following search terms: “elderly” OR “old*” OR “resident*” OR “aged” OR “nursing home residents” OR “restraint*” OR “restrictive practices” OR “physical restraint*”, “nursing home*” OR “long-term care*” OR “long-term care facilities” OR “type” OR “reason” OR predictors, OR “risks” OR “intervention”.

The database search was performed using EndNote software (version X9, Clarivate Analytics). As a result of the search, 871 studies were identified, of which 63 remained after all the titles and abstracts were assessed and duplicates were removed. PRISMA (Figure 1) shows the study selection process for inclusion in the review.

### 2.3. Inclusion and Exclusion Criteria

In the literature, restraint methods are classified as physical, mechanical, chemical, environmental, and seclusion [15]. In this review, physical restraint is defined as anything attached to a person’s body to restrict or control movement/behavior, or other physical barriers such as bed rails [45]. Therefore, those studies that define a physical restraint as something attached to a person’s body or as a physical barrier restricting movement or behavior were included in this scoping review. It is essential to understand that the criteria for determining “old age” can vary across nations, influenced by age-related demographics and cultural views. Different nations might define this age benchmark based on elements like life expectancy, societal roles, retirement norms, or health indicators. In this study, we did not specifically classify the age constituting older adults. The review sought to identify peer-reviewed primary studies concerning the prevalence and type of physical restraint used among elderly people living in nursing homes, factors affecting the use of physical restraint, and interventions employed to reduce physical restraint use. The scoping review searched for primary research studies published during 2000–2021 with a title and/or keywords that included “physical restraint” and/or “elderly in nursing homes”. Articles published in English were included. The excluded items were studies that did not sample elderly residents in nursing homes, were published in languages other than English, were published as review articles, were letters to the editor, conference papers, editorials, protocols, commentaries, or expert opinions, or that were impossible to retrieve as complete articles or book chapters.

### 2.4. Study Selection

Inclusion and exclusion criteria were used to select the studies selected. The selected abstracts were screened and read mainly by the two authors (GH, SK). The same authors independently reviewed the full text of the articles for inclusion, and any disagreements (eight full-text articles) were resolved through discussion until a consensus was achieved by all three researchers (GH, SK, DMP). It was decided that 28 of the 63 articles that did not meet the inclusion criteria should not be included. On completion of the review process, 35 articles were identified for charting. The selected articles were analyzed according to year, country, setting, sample, method, and three outcomes (methods and prevalence of physical restraint, reasons for using physical restraint, and interventions), as determined using previous literature.

### 2.5. Charting the Data

A data charting form was developed to facilitate the extraction of the author/year, title, research location, aim/purpose, method, prevalence, type of physical restraint used among the elderly residents of nursing homes, risk factors, and interventions to reduce physical restraint use. All the data from the charting form was imported into Excel for analysis. This review’s final stage consisted of a results summary and thematic analysis. The data in this scoping review were charted, analyzed and summarized by using the PAGER framework suggested by Bradburry-Jones et al. (2021) [46]. The PAGER framework was suggested to improve the quality of the review and guide future research about the advances and gaps related to the topic of the review. We specifically focused on the “Gap”, “Evidence”, and “Research” steps in this study. To visually enhance our presentation, we created a table detailing the gaps, evidence, and future research areas in our review.

## 3. Results

### 3.1. Characteristics of the Included Studies

This review retrieved 631 articles on physical restraint use among elderly nursing home residents via databases, together with 18 articles identified via a web search and a search of grey literature. Of the 649 retrieved articles, 35 met the inclusion criteria. Of those, the majority (11) had a cross-sectional design. Thirty of the articles were published between 2010 and 2022. The largest number of articles, 12, originated from Germany and the Netherlands, followed by the United States (4), Australia (2), Canada (3), China (3), and Norway (4). Two studies sampled residents from nursing homes in multiple countries. Sample sizes ranged from 264,068 to 5 residents, depending on the study’s design. Of the 35 included articles, 26 reported on the prevalence of physical restraint use, 26 studied methods of physical restraint, and 10 were studies on interventions to reduce physical restraints among elderly residents in nursing homes.

### 3.2. Prevalence and Types of Physical Restraint Use

The 26 studies providing data about the type and prevalence of physical restraint use originated from 22 countries. Two studies were multi-country, and the rest were single studies from Australia, Canada, China, Germany, the Netherlands, Norway, Singapore, Spain, Sweden, Switzerland, and the United States. The prevalence of physical restraint use was found to be highest in Spain (84.9%) and lowest in the USA (1.9%). A variety of restraint devices were reported in nursing homes, including bed rails, belts, trunk and chair restraints, pelvic straps in an armchair, tight bed sheets, side rails, fixed tables, vest restraints, sleep suits, bed belts, and many others. The most common device reported was bed rails, which appeared in 11 articles, with the highest prevalence in Singapore (98%) and the lowest (4.7%) in Germany. The second most commonly used method was chair restraint; the highest prevalence was reported to be 57%. Trunk use was reported in 5 studies, with the highest prevalence being 45%. Daily limb and/or trunk restraint use was reported to be 12% in Italy, 8% in Belgium, 4% in Finland, 1% in England, 0.4% in Poland, and 0% in the Netherlands. Limb restraints were the least popular; their use was reported to be 1.2% in Spain and 0.3% in China. The highest prevalence values of fixed table, belt restraint, belt in bed, sleeping suit and sheet in bed, and vest were 36%, 27%, 9.9%, 4%, and 6.1%, respectively. In one study, a pelvic strap in an armchair was reported to have been employed against elderly nursing home residents (Table 1).

### 3.3. Reasons for Physical Restraint Use

Of the 35 included articles, the 18 published between 2007 and 2020 reported the reasons for using physical restraint in Australia, Brazil, Canada, China, Finland, Israel, the Netherlands, Norway, Spain, Switzerland, and the United States. The following reasons were given for restraining the patients: age, care dependency, level of disablement, impaired activities of daily living, cognitive status, dementia, Alzheimer’s, Parkinson’s, negative mood, hallucinations, delusions, disorientation/confusion, depression, preventing dislodgement of feeding tubes, safe use of medical devices, workload, staff culture, location and availability of human resources, negative experiences of nurses, concerns and uncertainties of relatives and legal guardians, and organizational problems such as staff fluctuations and shortages of physicians. The largest number of studies (13) reported the prevention and/or risk of falls to be the main reason for using physical restraints, followed by behavioral problems such as wandering (7), verbal or physical agitation (6), being verbally or physically abusive (1), injury to others (1), shouting, restlessness, aggressiveness, disrobing in public, and resisting care (1), functional impairment (1), urinary or fecal incontinence (3), hip fracture/fall-related fractures (2), history of falls (1), bedfast (1), and being untidy (1). The risk of self-injury was reported in three studies (Finland, Israel, and Singapore) as being the reason for using restraints. One study in Germany reported on the necessity of restraining a patient due to polypharmacy (Table 2). 

### 3.4. Interventions to Reduce Physical Restraint Use

Interventions to reduce physical restraint use in nursing homes were identified in 10 studies: Germany (3), the Netherlands (4), Norway (1), Spain (1), and Sweden (1). Most studies reported guideline- and/or theory-based multicomponent interventions consisting of the training and education of nursing home staff, including nurses, licensed practical nurses, nurses’ aides, physicians, etc. Other interventions included institutional policy changes to discourage the use of physical restraints, consultation, surveillance technology, and small-scale living facilities. Some of the educational intervention studies showed that it is possible to eliminate or reduce physical restraint use among the elderly in nursing homes [35,40,54,55,58,60], while others reported just the opposite [47,52]. A study from Germany used an intervention to prevent behavioral symptoms and fall injuries by educating nursing home staff using a 6-h training course and technical aids, such as hip protectors and sensor mats [54]. A study from the Netherlands suggested the positive effects of small-scale, home-like facilities [63]. Another study suggested that surveillance technology would give residents with dementia more freedom of movement and should be considered before physical restraint [62] (Table 3).

### 3.5. The PAGER Analysis

The patterns, advances, gaps, evidence for practice and research recommendations are shown in Table 4. The results were summarized under four patterns. These included the prevalence, types of physical restraint, factors affecting physical restraint use, and interventions to reduce physical restraint use in elderly residents in nursing homes. This review showed that there is strong evidence supporting the substantial use of PR with a variety of physical restraint devices in nursing homes, particularly from studies in countries with large elderly populations. Studies on interventions focused on training and education programs for nursing home staff.

## 4. Discussion

This review study aimed to examine the prevalence and methods of physical restraint use among the elderly in nursing homes. It systematically describes the reasons for using physical restraint, and maps interventions and their effectiveness in avoiding its use in nursing homes. 

This review revealed the diversity of devices used to restrain the elderly in nursing homes. The prevalence of physical restraint varied widely across countries, ranging from 1.9% in the USA to 84.5% in Spain. Our findings aligned with the literature [42,66]. The Nursing Home Reform Act (1987), which gave nursing home residents the right to be free from restraints employed for disciplinary purposes or for the convenience of staff, helped reduce the rate of restraint use. This explains the low level of restraint use in the USA. It is well established in the literature that physical restraint remains common in nursing homes despite their lack of effectiveness or safety, and should be used only if there are no alternatives [19,35,52,67,68,69]. The most common arguments against restraint-free elderly care were debunked in the Editorial, “Zero tolerance for physical restraints: Difficult but not impossible”. The article argued that unrestrained care is possible and should be part of standard care for all older people [70]. Sadly, this scoping review revealed the continuing popularity of physical restraint. According to Marques and colleagues, the prevention of falls is often considered an indicator of quality of care [71]. Therefore, physical restraint may be used to decrease falls and thus make the nursing home staff look good.

This study found that bed rails are physical restraint device used most frequently among the elderly. Bed rails were more common in the Netherlands [29,35,52] and Spain [48] than in Germany [47] and Switzerland [41]. These findings are consistent with previous studies focusing on physical restraint use in nursing homes [20,28,45,66,71]. Although bed rails may seem harmless, and even help one to sleep better by eliminating the fear of falling out of bed, injuries can still occur, such as getting caught between the rails [19,72]. Recent studies have revealed that the use of bed rails is contraindicated because there is no scientific evidence supporting their effectiveness in preventing injuries among older adults [45,69,71,73]. Side rails are suggested only if the patient has freedom of movement and can exit the bed by removing the rails or if patients use them to reposition themselves [74]. The use of bed rails may also impair the dignity and autonomy of the elderly. Although the decision to impose bed rails is made by physicians, nurses, who are part of the decision-making process, are responsible for the implementation and control of physical restraint and protecting the rights of the elderly residents. Bed rails should only be used as a last resort, and even then with a doctor’s order; this is to avoid the loss of dignity, self-respect, self-confidence and self-esteem [71]. Bed rails should not be used for people with impaired cognitive function or dementia/Alzheimer’s disease without first considering alternative strategies, such as lowering the bed height [19,74].

Identifying reasons for using physical restraint among the elderly in nursing homes is crucial before considering interventions to replace them. Most of the studies in this review reported preventing falls as the main reason for using physical restraint. This finding is consistent with the literature [29,67,74]. Falls are common among the elderly due to age-related changes, such as impaired gait and balance, weak muscles, and impaired vision. The risk of falling increases after 60 years of age [75,76,77]. In contrast, studies have reported that fall prevention is not enough to justify physical restraint due to it being ineffective in reducing injuries [29,78,79]. Indeed, falls are one of the consequences of being physically restrained [22,77]. On the other hand, in the case of active older people with cognitive impairment, the risk of falling may increase [78]. Yet, other studies have provided strong evidence that fall rates can be reduced by resistance exercises with balance training and muscle strengthening in the lower extremities [48,80,81]. 

The second most common rationale for using physical restraints was patient-related factors such as wandering, agitation, and cognitive impairment. Similar results were seen in previous studies [14,42,49]. While the behaviors of the elderly may be a factor, studies have also shown that using physical restraint triggers aggressive behaviors in older people [82,83]. According to a new law in Switzerland, threatening injury to oneself or others is the only acceptable reason to physically restrain older persons [41]. 

This review revealed a few simple interventions to reduce physical restraint use among the elderly in nursing homes. Most of the interventions involved training and education programs for nursing home staff. Unfortunately, owing to the design and sample of the study, the effectiveness of the interventions was found to be inconclusive. The result was in line with recent studies on the same subject [61,84]. A study assessing the association between surveillance technology and restraint found no significant relationship [68]. A recent review reported that physical restraint-free care is possible by creating environments that meet the needs of older people with mobility and cognitive impairments, and that promote patient safety [85]. This finding is supported by Evans and Cotter [73], who stated that reducing the use of physical restraint depends on a multitude of interventions. Individuals with dementia can be managed without the use of restraints, no matter the environment, by creating a tailored care approach that addresses and anticipates their unique needs and behaviors. This approach can be complemented by organizational reforms. Additionally, tools like video monitoring and electronic alert systems offer added protection to prevent falls among the elderly [58,86]. According to O’Keeffe [69], preventing falls is possible by improving the education and guidance provided to staff. The author also called for further efforts to educate staff, informing them that using bed rails is an uncertain and insecure approach to preventing people from falling out of bed. Thus, there is a continuing need for effective interventions to reduce physical restraint use in elderly nursing home residents [17]. Strengths and Limitations

Several limitations of our study should be noted. First, the results of this scoping review were limited to the key search terms used in the research and focused on studies published between 2000 and 2021. Second, this study did not differentiate between studies with or without bed rails, which may have also affected the reporting of physical restraint use in nursing homes. Third, the authors may have missed some crucial evidence due to the limited language criteria. Fourth, we recognize that there is significant variance in the level of evidence across various study designs. However, in this review, we did not analyze the results based on the study design. Lastly, no quality assessment was performed on the articles. In contrast, the main strength of this review lies in its use of the PAGER framework, which guides future researchers regarding the advances and gaps present in the research, offers suggestions for future research regarding the topic of the review, and achieves consensus through discussions among researchers.

## 5. Conclusions

In conclusion, this review provides valuable insights into the use of physical restraints among elderly residents in nursing homes. Despite efforts to minimize their use, physical restraints continue to be employed, particularly with elderly individuals who have cognitive impairments. Patient-related factors such as wandering, agitation, and cognitive impairment were identified as the second most common reasons for using physical restraints in this population. To address this issue, it is crucial to enhance the skills of nursing home staff, especially nurses, so that they can provide safe and ethical care for elderly residents with cognitive and functional impairments, aggressive behaviors, and fall risks.

The ongoing use of physical restraints for fall prevention, despite lacking scientific evidence, raises ethical concerns. To address this, it is recommended that clear directives are provided to nursing home staff based on the latest evidence-based practices and interventions for fall prevention. These directives should emphasize the safety, rights, and dignity of the residents, ensuring that physical restraints are only considered as a last resort when no alternatives are available. Additionally, implementing mandatory training programs for nursing home staff is essential. This training should focus on recognizing and responding to the individual needs and behaviors of residents with cognitive impairments, emphasizing person-centered care approaches. Understanding and addressing the underlying causes of behaviors that lead to restraint use is crucial. Person-centered interventions and environmental modifications should be adopted to effectively address these causes. By creating an environment that supports individualized care plans and promotes autonomy and well-being, the need for physical restraints can be minimized.

Implementing these policy suggestions will enable nursing homes to create a safe, supportive, and person-centered environment that upholds the rights and dignity of elderly residents. A study by Laurin et al. suggested that interviewing nursing staff is a reliable method of data collection for measuring physical restraint use among residents [66]. Thus, further research is required on the experiences of nurses when managing such residents in order to guide the ethically and clinically challenging decision to use physical restraint.

## Figures and Tables

**Figure 1 healthcare-11-02204-f001:**
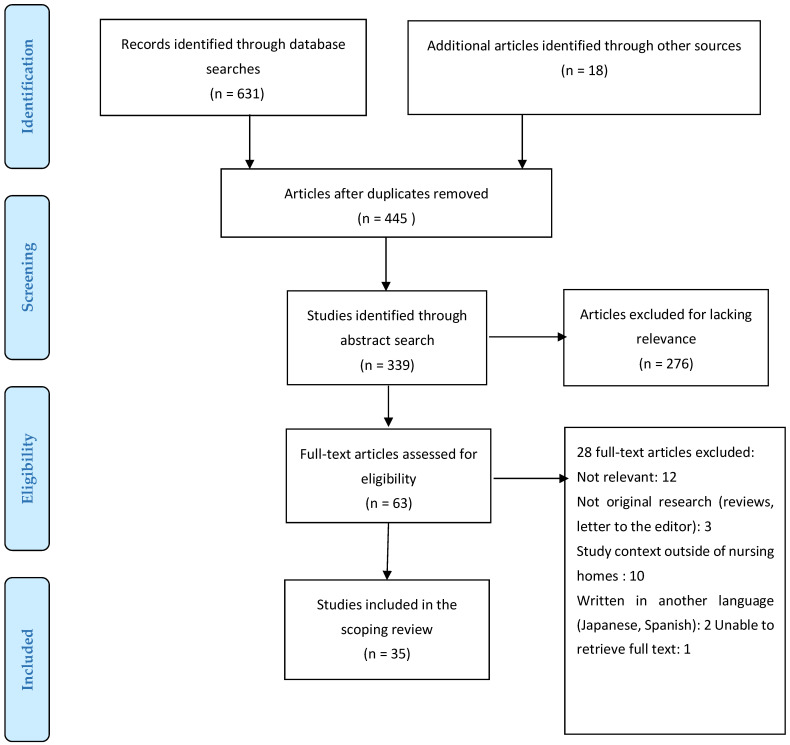
PRISMA Flowchart.

**Table 1 healthcare-11-02204-t001:** Type and prevalence of physical restraints used against elderly residents of nursing homes.

Author, Year Country	Aim/Purpose of Study	Design and Study Population	PR Use and Methods
Abraham et al., 2019 [47], Germany	To evaluate the effectiveness of two versions of a guideline and theory-based multicomponent intervention to reduce physical restraints in nursing homes.	RCT-120 nursing home residents The mean age: 82.5 years	PR: (Baseline) 18.6% Bed rails: 16.1% Any belt: 0.8% Belt in chair: 0.8% Fixed table: 1.1% Belt in bed: 0.1% Other: 3.0%
Aranda-Gallardo et al., 2018 [48], Spain	To determine the characteristics of a typical institutionalized elderly patient who suffers a fall and to describe the physical harms resulting from this event.	Prospective cohort, multi-center study—647 nursing home residents The mean age: 81.81 years	PR: 16.79% Bed rails: 53.53%
Bellenger et al., 2017 [16], Australia	To investigate the nature and extent of physical restraint deaths reported to Coroners in Australia over a 13-year period.	Retrospective cohort study— 58 nursing home residents placed under physical restraint. The median age: 83 years	PR: NI Chair restraint Bed rails Fixed table Cot sides with webbing
Delvalle et al., 2020 [49], Brazil	To estimate the prevalence of mechanical restraint in nursing homes and the factors associated with its performance.	Cross-sectional study— 443 elderly in 14 nursing homes The mean age: 83.8 years	PR: 7.45% Wheelchair: 27.3% Plastic chair: 18.2% Bandage: 3.0% Bed rail: 45.5% Adapted wooden rail: 3.0% Sheeting: 48.5%
Castle and Engberg, 2009 [21], USA	To examine whether physical restraint use contributes to subsequent physical or psychological health decline.	Longitudinal study—264,068 nursing home residents	PR: 1.9%
Estévez-Guerra et. al., 2017 [50], Spain	To examine the prevalence of PR use in long-term care residents with the ability to move voluntarily.	Cross-sectional observational and correlational multi-center study— 920 long-term care residents The mean age: 81 years	PR: 84.9% Side rails: 84.5% Belts in chair: 26.9% Belts in bed: 9.9% Chair with attached table: 6.2% Vest: 6.1% Wrist/ankle belt: 1.2% Sleep suits: 1.2%
Feng et al., 2009 [17], Canada, Finland, Hong Kong, Switzerland, and USA	To compare inter- and intra-country differences in the prevalence of PR and antipsychotic medications in nursing homes.	Population-based, cross-sectional study—14,504 residents of nursing homes The average age: 82–84 years	Switzerland: 6% The USA: 9% Hong Kong: 20% Finland: 28% Canada: 31%
Foebel et al., 2016 [24], Czech Republic, England, Finland, France, Germany, Israel, Italy, and the Netherlands	To explore antipsychotic medications and PR use and their effects on physical function and cognition in older nursing home residents.	Retrospective cohort study— 532 residents with dementia in 57 nursing homes The mean age: 85.2 years	PR: 19.6% Trunk: 45% Chair: 55%
Freeman et al., 2017 [25], Canada	To examine the role of physical restraint use, use of antipsychotic medications, and engagement in social activities in affecting change in cognitive status and driving cognitive decline among residents newly admitted to a Long term care facilities (LTCF).	Longitudinal Secondary data analysis— 111,052 residents in 635 LTCFs The mean age: 82.9	PR: 13.2% Trunk restraint: 8.5% Limb restraint: 0.3% Chair that prevents rising: 6.9%
Gulpers et al., 2011 [35], the Netherlands	To test the effectiveness of EXBELT on reducing belt restraint usage in psychogeriatric nursing home care.	A quasi-experimental longitudinal study—405 nursing home residents The mean age: 83.3 years	PR (Baseline): 61% Belts: 18% Wheelchair: 15% Bed: 6% Wheelchair with a locked table: 12% Special sheet: 9% Full-enclosure bedrails: 55.5% Chair on a board:1.5% Deep or overturned wheelchair: 8.5% Sleep suits: 7.5%
Hamers et al., 2004 [29], the Netherlands	To examine the prevalence of PR use in cognitively impaired nursing home residents, the manner in which restraints are used, the reasons for using them, and the relationships between residents’ characteristics and the use of PR.	A point prevalence study— 260 nursing home residents The mean age: 81 years	PR: 49% Bed and chair-26% Bed: 23% Bed rails: 98% Belts: 27% Belt with chairs: 57% Chairs with a table: 36%
Heckman et al., 2017 [12], Canada	To describe the clinical complexity of older institutionalized persons with Parkinson’s disease (PD); and examine patterns and predictors of restraint use and the prescription of antipsychotics in this population.	Cross-sectional cohort study— 7851 Complex Continuing Care (CCC) residents with a recorded diagnosis of PD The mean age: 82.6 years	PR: 18.9% Trunk: 11.3% Limb: 0.4% Chair: 11.6%
Heeren et al., 2014 [51], USA	To examine the relationship between staffing levels and the use of physical restraints in nursing homes.	Multi-center study—570 Residents in 23 wards in nursing homes (NHs) The median age: 86 years	PR: 47.5%
Heinze et al., 2012 [38], Germany	To investigate factors related to the use of restraints and to explore whether the number of nurses was an influencing factor regarding the use of restraints in German NHs and hospitals.	A secondary analysis of a cross-sectional study—5521 residents The mean age: 84.9 years	PR: 26.3% Bed rails: 25.77% Belts for fixation: 4.8%
Hofmann et al., 2015 [41], Switzerland	To investigate the prevalence and types of physical restraints used in nursing homes in two Swiss cantons and to explore whether resident-related and organizational factors are associated with the use of physical restraints.	A multi-center cross-sectional study—1362 residents The mean age: 85.1 years	PR: 26.8% Centre prevalence: 2.6% to 61.2% Bilateral bedrails: 20.3% Unilateral bedrails at one side of the bed with the other positioned at the wall: 5.7% Wheelchair with a locked tray table: 1.8% Belt in chair: 1.1% Sleep suits: 1.1% Chair preventing rising: 0.5% Chair with a locked tray table: 0.3%
Huizing et al., 2007 [37], the Netherlands	To investigate the relationship between the use of physical restraints on psycho-geriatric nursing home residents and the characteristics of organizations and residents.	Cross-sectional study— 371 residents The mean age: 83 years	PR: 56%
Huizing et al., 2009 [52], the Netherlands	To investigate the effects of an educational intervention on the use of physical restraints on psychogeriatric nursing home residents.	A cluster-randomized trial—371 residents in a psychogeriatric nursing home The mean age: 83 years	PR (Baseline): 51.5% Belt in chair: 10% Belt in bed: 9% Bilateral bedrails: 45% Deep or tipped chair: 18% Special sheet: 4% Sleep suits: 8% Sensor mat: 4% Infrared system: 4%
Kirkevold and Engedal, 2004 [53], Norway	To describe the prevalence of various types of constraint in Norwegian nursing homes.	Descriptive study—1501 residents in 222 nursing home special care units The mean age: 84.4 years	PR: 36.1% Bedrails without patients consent; 32.2% Belts or other fixing to bed: 2.3% Belts or other fixing to chair: 8.5% Other physical restraint: 3%
Koczy et al., 2011 [54], Germany	To evaluate the effectiveness of a multifactorial intervention to reduce the use of physical restraints in residents of nursing homes.	Cluster-randomized controlled trial—333 residents in 45 nursing homes	PR (Baseline): 6.1%
Köpke et al., 2012 [55], Germany	To reduce PR prevalence in nursing homes using a guideline- and theory-based multicomponent intervention.	Parallel group cluster RCT—2283 residents (IG), 2166 residents (CG) in 36 nursing homes	PR (Baseline): 31.1% Restrictive bed rails: 29.1% Any waist belt: 2.9% Waist belt in bed: 0.8% Waist belt in chair: 2.4% Fixed table: 1.9% Other physical restraint: 3.8%
Lam et al., 2017 [56], China	To review the change in the prevalence of physical and chemical restraint use in LTCFs over a period of 11 years in Hong Kong and to identify the major factors associated with their use.	Longitudinal Study—2896 Residents in 10 residential LTCFs Mean age: 83.3 years	PR: 70.2%
Luo et al., 2011 [27], USA	To estimate the use of different types of physical restraint and assess their association with falls and injuries among residents with and without AD or dementia in US nursing homes.	Cross-sectional study—5057 nursing home residents with Alzheimer Disease (AD) or dementia and 4224 residents without	PR: 6.99% Bed rails: 36.79% Limp: 0.40% Trunk: 3.87% Chair restraints: 3.35%
Mamun and Lim, 2005 [23], Singapore	To assess the use and complications related to physical restraints in Singapore nursing homes.	Mixed-method study—390 nursing homes residents The mean age: 80.1 years	PR: 23.3%
Meyer et al., 2009 [57], Germany	To investigate the prevalence of physical restraints, the frequency with which the devices being applied and the frequency with which psychoactive medication is available on demand during 12-month follow-up, and characteristics associated with restraint use in nursing homes.	Cross-sectional study—2367 nursing homes residents The mean age: 86 years	PR: 26.2% Bed rails: 24.5% Waist belt used in a chair or bed: 2.7% Chair with a table: 2.1% Other devices: 2.3%
Muniz et al., 2016 [40], Spain	To implement a dementia-friendly culture as well as specific organizational skills relevant to person-centered care and environmental improvement.	Longitudinal study 4361, 2410 residents with dementia—41 Spanish nursing homes Mean age: 84.6	PR: 18.1% and 29.1% with dementia Bed rails: 43.5% and 56.8% Chair abdominal belt: 9.4% and 15.1% Upper body vest and perineal belt: 3.4% and 5.7% Belt in bed: 5.9% and 9.8% Wrist restraint: 0.8–1.4%
Pellfolk et al., 2010 [58], Sweden	To evaluate the effects of a restraint minimization education program on staff knowledge, attitudes and use of PRs.	Cluster RCT—184 staff and 191 residents (IG), and 162 staff and 162 residents (CG) in dwelling units for people with dementia The mean age: 43.5 years	PR (Baseline): 25.2%
Pivodic et al., 2020 [59], Belgium, England, Finland, Italy, the Netherlands and Poland	To determine the frequency of physical limb and/or trunk restraint use in the last week of nursing home residents’ lives in six European countries and its association with country, resident and nursing home characteristics.	Epidemiological cross-sectional survey study—1384 deceased residents from 322 nursing homes The mean age: 83–89 years	Belgium: 8% England: 1% Finland: 4% Italy:12% Poland: 0.4%
Testad et al., 2016 [60], Norway	To evaluate the effect of a tailored 7-month training intervention, entitled “Trust Before Restraint”, on reducing the use of restraint, agitation, and antipsychotic medications in care home residents with dementia.	RCT—274 residents with dementia in 24 care homes. The mean age: 88.2 years	PR: 12.4% Bedrails without the patient’s consent: 4.7% Belts or other fixing to bed: 0.4% Belts or other fixing to chair: 0.4% Physical retention: 2.9%
Wang et al., 2022 [61], China	To identify the relationship between the Theory of Planned Behavior constructs and nursing staffs’ use of PR in LTCFs.	Cross-sectional survey—316 nursing staff in six Chinese LTCFs The mean age: 43.52 years	PR: 25.83%
te Boekhorst et al., 2013 [62], the Netherlands	To explore the social, mood and behavioral dimensions of the quality of life of residents under surveillance technology compared with those of residents under PR.	An explanatory study—150 nursing home residents	PR: 5.4% Fixation: 52% Restrictive chair: 48%
Verbeek et al., 2014 [63], the Netherlands	To examine the effects of small-scale living facilities on the behavior of residents with dementia and the use of physical restraints and psychotropic drugs.	A quasi-experimental study—259 nursing home residents with AD or dementia The mean age: 82.4 years	PR: 44% Belt: 11% (Wheel) chair with a locked table/chair on a board: 10% Deep or overturned chair: 8% Bilateral full enclosed bed rails: 40% Sleepsuits: 16%

PR: Physical restraint; LTCF: Long-term care facilities; AD: Alzheimer’s disease; RCT: Randomized controlled trial; IG: Intervention group; CG: Control group.

**Table 2 healthcare-11-02204-t002:** Reasons given for using physical restraint.

Author, Year, Country	Aim and Purpose of Study	Design and Study Population	Reasons
Bellenger et al., 2017 [16], Australia	To investigate the nature and extent of physical restraint deaths reported to coroners in Australia over a 13-year period.	Retrospective cohort study: 58 nursing home residents experiencing physical restraint	Impaired mobility, dementia, risk of fall, history of repeated falls, mobility
Ben Natan et al., 2010 [64], Israel	To identify and analyze major variables affecting the decision of nursing staff to physically restrain elder residents of long-term care facilities.	Descriptive correlational study: 10 4 nurses in a geriatric care institution	Dementia, physical state, stress of elder residents, cognitive impairment: 10% Risk of fall: 53.8% Risk of self-injury: 80.8% Threatening the lives of others: 66.3%
Delvalle et al., 2020 [49], Brazil	To estimate the prevalence of mechanical restraints in nursing homes and factors associated with their performance.	Cross-sectional study: 443 elderly in 14 nursing homes	Risk of falls: 66.7% Agitation, aggressiveness, wandering: 21.2% Lack of institutional protocol or medical request: 9.1% Alzheimer’s Disease: 3.0%
Estévez-Guerra et al., 2017 [50], Spain	To examine the prevalence of physical restraint on long-term care residents with the ability to move voluntarily.	Cross-sectional observational and correlational multi-center study: 920 long-term care residents	Prevent falls, impaired cognitive status
Feng et al., 2009 [17], Canada, Finland, Hong Kong, Switzerland, and USA	To compare inter- and intra-country differences in the prevalence of physical restraint and antipsychotic medications in nursing homes.	Population-based, cross-sectional study: 14,504 residents of nursing homes	Longer resident stays were associated with higher physical restraint use rate in Switzerland Larger facilities had a lower rate of physical restraint use in Canada and Finland Older age was associated with reduced physical restraint use only in the USA.
Foebel et al., 2016 [24], Czech Republic, England, Finland, France, Germany, Israel, Italy, and the Netherlands	To explore antipsychotic medications and physical restraint use and their effects on physical function and cognition in older nursing home residents.	Retrospective cohort study: 532 residents with dementia in 57 nursing homes	Dependent: 55.8% Incontinence: 97.1% Severe CI: 59.6% Hallucinations: 14.4% Delusions: 14.4% Wandering: 24.0% Disrobing in public: 34.6% Verbally abusive: 27.9% Physically abusive: 16.4% Socially inappropriate: 30.8% Resist care: 4.8% History of falls: 12.5%
Hamers et al., 2004 [29], the Netherlands	To examine the prevalence of physical restraint use in cognitively impaired nursing home residents, the manner in which restraints are used, reasons for using them, and relationships between residents’ characteristics and the use of physical restraint.	A point prevalence study: 260 nursing home residents	Prevent falls: 80% Restlessness: 24% Safe use of medical devices: 1% Poor mobility Care dependency Risk of falling in the opinion of nursing staff
Heckman et al., 2017 [12], Canada	To describe the clinical complexity of older institutionalized persons with PD, and examine patterns and predictors of restraint use and prescription of antipsychotics in this population.	Cross-sectional cohort study: 7851 Complex Continuing Care (CCC) residents with a recorded diagnosis of PD	History of falls Cognitive impairment Aggressive behavior Delusions or hallucinations Behavioral symptoms Functional impairment Urinary incontinence Shortage of physicians
Heeren et al., 2014 [51], USA	To examine the relationship between staffing levels and the use of physical restraints in nursing homes.	Multi-center study: 570 residents, 23 wards in 7 nursing homes	Bathing dependency Transfer difficulties Risk for falls Frequent restlessness/agitation
Heinze et al., 2012 [38], Germany	To investigate factors related to the use of restraints and to explore whether the number of nurses is an influencing factor regarding the use of restraints in German nursing homes and hospitals.	A secondary analysis of a cross-sectional study: 5521 residents	Care dependency Impaired mobility Bedfast Urinary incontinence History of falls Polypharmacy High risk of falls Disorientation/confusion
Hofmann et al., 2015 [41], Switzerland	To investigate the prevalence and types of physical restraint used in nursing homes in two Swiss cantons and to explore whether resident-related and organizational factors are associated with the use of physical restraints.	A multi-center cross-sectional study: 1362 residents	Age Degree of care dependency Mobility limitation Verbal agitation Physical agitation Risk/history of fall and/or fracture
Huizing et al., 2007 [37], the Netherlands	To investigate the relationship between the use of physical restraints with psycho-geriatric nursing home residents and the characteristics of organizations and residents.	Cross-sectional study: 371 residents	Age: 84.0% Female: 78.6% Cognitive status: 4.5% ADL: 4.6% Nursing staff workload: 3.1% Higher job autonomy (nursing staff): 3.0% Immobility: 3.0%
Koczy et al., 2011 [54], Germany	To evaluate the effectiveness of a multifactorial intervention to reduce the use of physical restraints on residents of nursing homes.	Cluster-randomized controlled trial: 333 residents in 45 nursing homes	Limited physical mobility Female High need for care
Köpke et al., 2012 [55], Germany	To reduce physical restraint prevalence in nursing homes using a guideline- and theory-based multicomponent intervention.	Parallel group cluster RCT: 2283 residents (IG), 2166 residents (CG) in 36 nursing homes	Negative experiences of nurses Concerns and uncertainties of relatives and legal guardians Organizational problems (e.g., staff fluctuation)
Lam et al., 2017 [56], China	To review the change in the prevalence of physical and chemical restraint use in LTCFs over a period of 11 years in Hong Kong and to identify the major factors associated with their use.	Longitudinal study: 2896 residents in 10 residential LTCFs	Impaired activities of daily living Impaired cognitive function Negative mood Bowel and bladder incontinence Dementia
Mamun and Lim, 2005 [23], Singapore	To assess the use and complications related to the use of physical restraints in Singapore nursing homes.	Mixed method study: 390 nursing home residents	Dementia Prevent falls: 18.7% Prevent dislodgement of feeding tubes: 22% Injury to self: 31.2% Injury to others: 8.6% Wandering: 23.7% Shouting: 36.6% Agitation: 8.8%
Meyer et al., 2009 [57], Germany	To investigate the prevalence of physical restraints, the frequency with which the devices areapplied, and the frequency with which psychoactive medication is available on demand during 12-month follow-up, and characteristics associated with restraint use in nursing homes.	Cross-sectional study: 2367 nursing homes residents	Degree of disablement Cognitive impairment Fracture Repeated verbal agitation
Øye et al., 2017 [32], Norway	To investigate what kind of restraint is used in three nursing homes and to investigate how staff use restraint under different nursing home contexts by comparing three nursing home settings.	Mixed-method study: 38 nursing home staff	Resident mix Staff culture Location Human resources Agitation Aggressiveness Wandering
Saarnio and Isola, 2010 [65], Finland	To describe the perceptions of nursing staff regarding the use of physical restraint in the institutional care of older people.	Qualitative study: 21 nursing home nurses	Requests by the patient’s family Aggressive Untidy Exposes him/herself Wandering Lack of legislation
Wang et al., 2022 [61], China	To identify the relationship between the theory of planned behavior (TPB) constructs and the nursing staff’s use of physical restraint in LTCFs.	Cross-sectional survey: 316 nursing staff in 6 Chinese LTCFs	Prevent falls Residents with feeding tubes

CG: Control group; CI: Cognitive impairment; IG: Intervention group; LTCF: Long-term care facility; RCT: Randomized controlled trial.

**Table 3 healthcare-11-02204-t003:** Summary of interventions to reduce physical restraints, their effectiveness and challenges.

Author, Year, Country	Aims and Objectives	Design and Study Population	Intervention	Effectiveness	Challenges and Suggestions
Abraham et al., 2019 [47], Germany	To evaluate the effectiveness of two versions of a guideline and theory-based multicomponent intervention to reduce physical restraints in nursing homes. The authors conducted a pragmatic cluster randomized controlled trial with a twelve-month follow-up including 120 nursing homes.	A pragmatic cluster RCT with 12,245 residents (4126 and 3547 in IG 1 and 2 and 4572 in CG)	Guideline-based multicomponent intervention. IG 1 received an updated version of a successfully tested guideline-based multicomponent intervention (comprising brief education for the nursing staff, intensive training of nominated key nurses in each cluster, introduction of a least-restraint policy and supportive material); IG 2 received a concise version of the original program, and the control group received optimized usual care (I.e., supportive materials only).	Neither intervention showed a clear advantage compared to the control.	Increased heterogeneity data, risk of bias in randomization. Other approaches, like legal or governmental policies, seem to be necessary to sustainably change physical restraint practice and reduce center variations in nursing homes.
Gulpers et al., 2011 [35], the Netherlands	To test the effectiveness of EXBELT on reducing belt restraint usage in psychogeriatric nursing home care.	A quasi-experimental longitudinal design with 714 residents in a nursing home	The intervention program included four major components: promotion of institutional policy change that discourages use of belt restraint, nursing home staff education, consultation by a nurse specialist aimed at nursing home staff, and availability of alternative interventions.	The intervention led to a substantial reduction in use of belts, full-enclosure bed rails, and sleep suits without increasing the use of other physical restraints, psychoactive drugs, or falls and fall related injuries.	Further research is recommended.
Huizing et al., 2009 [52], the Netherlands	To investigate the effects of an educational intervention on the use of physical restraints with psychogeriatric nursing home residents.	A cluster randomized trial of 371 residents in a psychogeriatric nursing home	The intervention consisted of five 2-h educational sessions for selected staff delivered over a 2-month period, one 90-min plenary session for all staff, and consultation with a nurse specialist (RN level). The educational program was designed to encourage nursing staff to adopt a restraint-free care philosophy and familiarize themselves with individualized care techniques.	No effect was had on the restraint status, restraint intensity, or multiple restraint usage in any of the three post-intervention measurements.	No information about factors potentially influencing the intervention’s lack of effectiveness due to contamination bias, insufficient consultation (one nurse specialist performed consultation). Future studies should include an effect evaluation and a process evaluation.
Koczy et al., 2011 [54], Germany	To evaluate the effectiveness of a multifactorial intervention to reduce the use of physical restraints in residents of nursing homes.	Cluster-randomized controlled trial with 333 residents	Persons responsible for the intervention in the nursing homes attended a 6-h training course that included education about the reasons for restraint use, its adverse effects, and alternatives to its use. The following components were selected: increasing awareness, improving knowledge, clarifying legal arguments, demonstrating alternatives, providing related equipment and supplies, and empowering staff members to participate in the decision-making process. Technical aids, such as hip protectors and sensor mats, were provided. The training was designed to give the change agents tools for problem-solving to prevent behavioral symptoms and injuries from falls without using physical restraints.	The intervention reduced restraint use (belts tied to a chair or to bed and chairs with fixed tables) without a significant increase in falling, behavioral symptoms, or medication.	Unblinded documentation of physical restraints and falls by the staff members of the nursing homes. An interdisciplinary approach based on medical and nursing science including ethical and legal aspects is likely to yield the greatest benefits.
Köpke et al., 2012 [55], Germany	To reduce physical restraint prevalence in nursing homes using a guideline- and theory-based multicomponent intervention.	Parallel group cluster RCT with 2283 (IG), 2166 (CG) residents of nursing homes	A guideline- and theory-based multicomponent intervention. Components were group sessions for all nursing staff; additional training for nominated key nurses; and supportive material for nurses, residents, relatives, and legal guardians. Control group clusters received standard information. As opposed to other guideline-based interventions, the central recommendation is not to perform a certain action, i.e., not to apply physical restraints, aiming to implement a “practice culture” without physical restraints.	A guideline- and theory-based multicomponent intervention compared with standard information reduced physical restraint use (bilateral bed rails, belts, fixed tables, and other measures limiting free body movement) without significant differences in falls, fall-related fractures, or psychotropic medication prescriptions.	Information leakage between head nurses and staff nurses who performed the assessment of the use of physical restraint.
Muniz et al., 2016 [40], Spain	To implement a dementia-friendly culture, as well as specific organizational skills relevant to person-centered care and environmental improvement.	A two-wave longitudinal study with 4361 residents with dementia	The intervention was initiated in April 2010 and combined training, consultation, and consultancy at various levels of the organization. “Dementia champions” (1 per nursing home) received in-depth training about a wide array of dementia topics (e.g., biological basis of dementia, genesis and management of behavioral and psychological symptoms of dementia, person-centered care) needed to coordinate the implementation of several dementia care approaches.	Physical restraints can almost be eliminated along with psychotropic medication.	Recliner chairs used as physical restraints were not tracked in the database. Future research is recommended to address ways to avoid injurious falls in restraint-free nursing homes for people with severe dementia.
Pellfolk et al., 2010 [58], Sweden	To evaluate the effects of a restraint minimization education program on staff knowledge, attitudes and use of physical restraints.	Cluster RCT with 156 staff and 185 residents (IG) and 133 staff and 165 residents (CG)	The intervention consisted of a six-month education program comprised of six different themes, one for each month, for nursing staff. The education included 30 min of videotaped lectures. Three of the lectures also included a clinical vignette presented in writing, which could be used for group discussions.	Staff education in small group dwellings can increase knowledge, change attitudes, and reduce the use of physical restraints in the care of elderly people suffering from dementia, without increasing the incidence of falls or the use of psychoactive drugs.	The effects In this study were measured immediately after the completion of the intervention. Thus, the long-term effects of the intervention cannot be evaluated based on this study; more studies are recommended.
te Boekhorst et al., 2013 [62], the Netherlands	To explore the social, mood, and behavioral dimensions of the quality of life of residents under surveillance technology compared with those of residents under physical restraints.	An explorative study with 150 residents	The use of surveillance technology versus physical restraints. Surveillance cameras, acoustic monitoring systems, chips worn in clothing or shoes that close doors or sound an alarm when off-limits doors are opened, tracking chips with GPS, inactivity sensors, movement sensors in beds or chairs, door sensors, and bed pressure sensors were defined as surveillance technology.	Not effective. Surveillance technology may only benefit those who can already move without the help of others.	There was no measurement made before residents were put under surveillance technology or physical restraint so as to establish potential baseline differences between these two groups. More robust design research is needed with surveillance technology.
Testad et al., 2016 [60], Norway	To evaluate the effect of a tailored 7-month training intervention, entitled “Trust Before Restraint”, on reducing the use of restraints, agitation, and antipsychotic medications in care home residents with dementia.	RCT with 197 residents with dementia and 35 staff	The intervention included the seven-step guidance group, where the care staff chose a situation includingthe use of restraint and the DMP model (to emphasize and understand the relationship between resident and care staff, and to support the identification and effective management of unmet needs in order to reduce the use of restraint and improve care).	Training intervention reduced the use of restraints in both the intervention and control groups, with a greater reduction in the control group.	Possible bias between the two groups that may have influenced the main findings of the study.
Verbeek et al., 2014 [63], the Netherlands	To examine the effects of small-scale living facilities on the behavior of residents with dementia and the use of physical restraints and psychotropic drugs.	A quasi-experimental study of 124 (IG) and 135 (CG) nursing home residents with AD or dementia	Comparing residents in two types of long-term institutional nursing care: small-scale living facilities versus traditional psychogeriatric wards on three time points—at baseline and at follow-ups after six and 12 months.	Positive effects of small-scale living facilities on the use of physical restraints and psychotropic drugs	Sample bias (residents were not randomized in a dementia care facility), risk of underreporting physical restraints (measurement of physical restraints was based on nurses’ self-reports instead of independent observers). Studies need to determine which elements of small-scale living facilities are essential for improving outcome measures and how they work together.

CG: Control group; IG: Intervention group; RCT: Randomized Controlled Trial.

**Table 4 healthcare-11-02204-t004:** Summary of the review according to the PAGER framework.

Patterns	Advances	Gaps	Evidence for Practice	Research Recommendations
Prevalence of Physical Restraint	Strong evidence supports the substantial use of PR in nursing homes, particularly from studies in countries with large elderly populations. Discussion of the possibility of creating physical restraint-free nursing homes.	Lack of extensive observational studies on the prevalence of physical restraints in long-term care facilities. Lack of understanding of the variation in physical restraint use rates among countries.	Physical restraint use indicates poor clinical practice and should be avoided.	Research is needed to explore the reasons for using physical restraint for performing daily activities. Future research should address interventions to avoid falls to reduce physical restraint use, especially for people with cognitive impairment/dementia.
Type of PRs	A variety of physical restraint devices are used in nursing homes among elderly populations. Studies about the everyday use of bed rails and chair restraints among the elderly.	Lack of research on the reasons for using different types of physical restraint and their effects on elderly residents’ behavior.	Evidence from future research is needed.	Further investigation is required to understand whether physical restraints are associated with various health outcomes.
Factors affecting PR use	The largest number of studies reported prevention and/or fall risk as the main reason for using physical restraints, followed by challenging behaviors.	Research has only focused on patient-related reasons for using PR among the elderly in nursing homes.	Improving the skills of nursing home staff, especially nurses, to care for the elderly with cognitive and functional impairment, aggressive behavior, and fall risk is warranted to eliminate physical restraint use in nursing homes.	Further research is needed to understand the variation in physical restraint use rates among countries; it is essential to determine the individual factors specific to each country.

## Data Availability

Data sharing not applicable.

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
