# Peer review of "Physical Restraint Use in Nursing Homes—Regional Variances and Ethical Considerations: A Scoping Review of Empirical Studies"

_healthcare, 2023, doi:10.3390/healthcare11152204_

Round 1

Reviewer 1 Report

There are some minor grammatical errors.

The study is very well grounded with extensive literature review.

The research methodology is very well discussed and in detail.

Not sure what it is means on page 13, line 283 "(Error! Reference source not found.)."  Please clarify.

Overall the content is excellent of the paper.  

Very well cited.

Well organized and written.

There are some minor grammatical errors.

The study is very well grounded with extensive literature review.

The research methodology is very well discussed and in detail.

Not sure what it is means on page 13, line 283 "(Error! Reference source not found.)."  Please clarify.

Overall the content is excellent of the paper.  

Very well cited.

Well organized and written.

Author Response

The study is very well grounded with extensive literature review.

The research methodology is very well discussed and in detail.

Overall, the content is excellent of the paper. 

Very well cited.

Well organized and written.

We would like to thank to the reviewer for his/her kind comments.

There are some minor grammatical errors.

Thank you for the comment. The manuscript was checked by grammatical errors and necessary corrections were made.  

Not sure what it is means on page 13, line 283 "(Error! Reference source not found.)."  Please clarify.

The necessary corrections were made throughout the manuscript.

Reviewer 2 Report

This is a very meaningful paper, there are some suggestions for the authors.

1. The method described by Joanna Briggs was followed to conduct a scoping review 153 without a quality assessment of the selected studies. Please add a citation and briefly introduce this research method.

2. The data in this scoping review were charted, analyzed and summarized by using PAGER  framework suggested by Bradburry-jones et al. (2021). Please explain how this paper applies this framework.

3.  It is possible to avoid restraint use in persons with dementia, regardless of setting, by developing an individualized plan of care to understand, prevent, and respond to patients’ needs and behaviors and organizational change. Observation and surveillance, such as video monitoring and electronic warning devices, are other possible interventions to protect the elderly from falls. The above statement has been discussed in many papers, and it is recommended to cite and discuss it more extensively. 

4.  Falls are common among the elderly due to age-related changes such as impaired gait and balance, weak muscles, and impaired vision. The risk of falling increases after 60 years of age (Rubenstein, 2006; Schoberer et al., 2022).  The authors should explore the causes of falls in elderly people in long-term care institutions and suggest rewriting this paragraph.

Author Response

This is a very meaningful paper, there are some suggestions for the authors

We would like to thank the reviewer for his/her kind comments.

The method described by Joanna Briggs was followed to conduct a scoping review 153 without a quality assessment of the selected studies. Please add a citation and briefly introduce this research method.

Thank you for pointing this out. We agree with the reviewer’s comment. The suggested changes were made, the paragraph was cited, and the necessary information was added.

Peters, M., et al., Guidance for conducting systematic scoping reviews. International journal of evidence-based healthcare, 2015, 13(3), 141–146. https://doi.org/10.1097/XEB.0000000000000050

The data in this scoping review were charted, analyzed and summarized by using PAGER  framework suggested by Bradburry-jones et al. (2021). Please explain how this paper applies this framework.

Thank you for pointing this out. We agree with the reviewer’s comment.

For this study, we adapted the PAGER framework by Bradburry-Jones et al. (2021) to fit the scope and objectives of our research. Specifically, we focused on the "Gap," "Evidence," and "Research" steps, as they were most pertinent to our research question and approach.

The data in this scoping review were charted, analyzed and summarized by using PAGER framework suggested by Bradburry-jones et al. (2021) [46]. The pager framework was suggested to improve the quality of the review and guide future researchers about the advances, gaps, and future research suggestions regarding the topic of the review. We specifically focused on the "Gap," "Evidence," and "Research" steps in this study. To visually enhance our presentation, we created a table detailing gaps, evidence, and future research areas to our review.

It is possible to avoid restraint use in persons with dementia, regardless of setting, by developing an individualized plan of care to understand, prevent, and respond to patients’ needs and behaviors and organizational change. Observation and surveillance, such as video monitoring and electronic warning devices, are other possible interventions to protect the elderly from falls. The above statement has been discussed in many papers, and it is recommended to cite and discuss it more extensively.

Thank you for pointing this out. The paragraph was rewritten and cited as suggested.

Individuals with dementia can be managed without the use of restraints, no matter the environment, by creating a tailored care approach that addresses and anticipates their unique needs and behaviors. This approach can be complemented by organizational reforms. Additionally, tools like video monitoring and electronic alert systems offer added protection to prevent falls among the elderly [49,78].

Falls are common among the elderly due to age-related changes such as impaired gait and balance, weak muscles, and impaired vision. The risk of falling increases after 60 years of age (Rubenstein, 2006; Schoberer et al., 2022).  The authors should explore the causes of falls in elderly people in long-term care institutions and suggest rewriting this paragraph.

Thank you for pointing this out. Our study specifically focused on physical restraint use among the elderly population in nursing homes.   

Reviewer 3 Report

I read your article with great interest. However, I am concerned about the following points, which led me to this opinion.

The reason for the choice of SEARCH TERMS is unclear. I believe that the bias of the results obtained in this REVIEW will be undeniable. I also infer that the results would be different if the search included words such as "dementia," "cognitive impairment," and "BPSD. These points need to be clarified.

The World Health Organization's definition of older adults is 65 years and older. However, the countries presented in the results do not necessarily define 65 as older adults. This point needs to be clarified before discussing the results.

I believe that the discussion will be clearer if the results obtained are examined by study design. For example, the level of evidence differs greatly between "RCT," "Cohort study," and "Cross-sectional study.

Author Response

I read your article with great interest. However, I am concerned about the following points, which led me to this opinion.

Thank you for your constructive feedback on our manuscript.

The reason for the choice of SEARCH TERMS is unclear. I believe that the bias of the results obtained in this REVIEW will be undeniable. I also infer that the results would be different if the search included words such as "dementia," "cognitive impairment," and "BPSD. These points need to be clarified

Thank you for pointing this out.

We view this as an opportunity for future research, where a more focused review could delve deeper into the relationship between cognitive impairments and physical restraint use. We believe that by presenting the current study as it is, we are paving the way for a more specialized exploration in subsequent studies, as suggested by your feedback.

In this review, the search strategy was formulated in alignment with the study's objectives and was reviewed by the librarian. Adjustments were made to address the raised issue in the manuscript and highlighted as shown below.

Keywords and MeSH terms were identified through an initial exploration of Web of Science and Pubmed. The terms were reviewed and agreed by the research team and the librarian from Izmir Tinaztepe University. After agreeing on the search terms, an electronic search was conducted to find eligible empirical articles using MEDLINE, PsycINFO, EMBASE, Web of Science, Scopus, Google Scholar, CINAHL, and grey literature by the librarian. The term “physical restraint” covered all restraints and was not differentiated as with or without a bed rail (bilateral/unilateral). The search strategy included the following search terms:

The World Health Organization's definition of older adults is 65 years and older. However, the countries presented in the results do not necessarily define 65 as older adults. This point needs to be clarified before discussing the results.

The necessary clarification was added to the method section of the manuscript, as shown below.

It is essential to understand that the criteria for determining "old age" can vary across nations, influenced by age-related demographics and cultural views. Different nations might define this age benchmark based on elements like life expectancy, societal roles, retirement norms, or health indicators. In this study, we did not specifically classify the age constituting older adults

I believe that the discussion will be clearer if the results obtained are examined by study design. For example, the level of evidence differs greatly between "RCT," "Cohort study," and "Cross-sectional study.

We appreciate your feedback, and we believe that a future review segmented by study design could offer significant insights. In this study, we adopted a methodology aimed at capturing the full scope of the literature on the subject. This enabled us to pinpoint major themes, uncover gaps, and determine potential areas for future research. The comment was acknowledged as a limitation in our review. Necessary changes were made in the manuscript as shown below.

Fourth, we recognize that there is a significant variance in the level of evidence across various study designs. However, in this review, we did not analyze the results based on study design. 

Round 2

Reviewer 3 Report

Thank you for your careful revisions.

I have the impression that it is easier to read than last time.

I would like you to correct the following

The layout of Table is difficult to read.

I believe this needs to be corrected to avoid confusion among readers.

Words such as p. 2 L48 elderly persons and p. 2 L83 older people are mixed up. They need to be unified.

Author Response

Thank you for your comments.

Page 2 Line 83, was changed as “elderly”.

The tables are fixed so that it is easy to read.

Kind regards,